**Data Availability Statement:** All relevant data are within the manuscript.

**Funding:** The work was funded through National Institutes of Health, NIH AI R01 AI132680 (to DC).

# Immobilization of Proteinase K for urine pretreatment to improve diagnostic accuracy of active tuberculosis

**Yosita Panraksa**[1][☯], **Anita G. Amin**[1][☯], **Barbara Graham**[2], **Charles S. Henry**[3,4], **Delphi Chatterjee**[1]*

**1** Mycobacteria Research Laboratories, Department of Microbiology, Immunology and Pathology, Colorado State University, Fort Collins, CO, United States of America, **2** Department of Microbiology, Immunology and Pathology, Colorado State University, Fort Collins, CO, United States of America, **3** Department of Chemistry, Colorado State University, Fort Collins, CO, United States of America, **4** School of Biomedical Engineering, Colorado State University, Fort Collins, CO, United States of America

☯ These authors contributed equally to this work.

* delphi.chatterjee@colostate.edu

## Abstract

The World Health Organization (WHO) calls for the development of a rapid, biomarker-based, non-sputum test capable of detecting all forms of tuberculosis (TB) at the point-of-care to enable immediate treatment initiation. Lipoarabinomannan (LAM) is the only WHO-endorsed TB biomarker that can be detected in urine, an easily collected sample matrix. For obtaining optimal sensitivity, we and others have shown that some form of sample pretreatment is necessary to remove background from patient urine samples. A number of systems are paper-based often destined for resource limited settings. Our current work presents incorporation of one such sample pretreatment, proteinase K (ProK) immobilized on paper (IPK) and test its performance in comparison to standard proteinase K (SPK) treatment that involves addition and deactivation at high temperature prior to performing a capture ELISA. Herein, a simple and economical method was developed for using ProK immobilized strips to pretreat urine samples. Simplification and cost reduction of the proposed pretreatment strip were achieved by using Whatman no.1 paper and by minimizing the concentration of ProK (an expensive but necessary reagent) used to pretreat the clinical samples prior to ELISA. To test the applicability of IPK, capture ELISA was carried out on either LAM-spiked urine or the clinical samples after pretreatment with ProK at 400 µg/mL for 30 minutes at room temperature. The optimal conditions and stability of the IPK were tested and validation was performed on a set of 25 previously analyzed archived clinical urine samples with known TB and HIV status. The results of IPK and SPK treated samples were in agreement showing that the urine LAM test currently under development has the potential to reach adult and pediatric patients regardless of HIV status or site of infection, and to facilitate global TB control to improve assay performance and ultimately treatment outcomes.

The funders had no role in study design, data collection and analysis, decision to publish, or preparation of the manuscript.

**Competing interests:** The authors have declared that no competing interests exist.

## Introduction

According to the recent WHO report, globally, 7.1 million people with tuberculosis (TB) were reported to have been diagnosed in 2019 –a small increase from 7.0 million in 2018 but a large increase from 6.4 million in 2017 and 5.7–5.8 million annually in the period 2009–2012 [1]. In 2020, the COVID-19 pandemic has already had a negative impact on access to TB diagnosis and treatment and will continue beyond 2021 [2].

While an estimated quarter of the world is latently infected with *Mycobacterium tuberculosis* (Mtb), active TB is caused by uncontrolled infection leading to a predominantly respiratory and transmissible disease. To fight against TB, better vaccines, therapies, and new tools for both diagnosis and research are critical.

Mtb has a unique cell wall with multiple lipid-based molecules that create a thick impermeable 'waxy' surface [3]. A major component of this cell envelope is lipoarabinomannan (LAM), which represents up to 15% of the bacterial mass [4–9] (**Fig 1A**). LAM is firmly but non-covalently attached to the inner membrane and extends to the exterior of the cell wall [10] where it interacts as a potent virulence factor that modulates the host immune response and plays an important role in the pathogenesis of Mtb infection [11].

When purified from laboratory culture, LAM's average molecular weight is 15 to17 kDa and runs as a smear on SDS/PAGE. The molecule is heterogeneous in size (**Fig 1B**), branching pattern, acylation, and phosphorylation on the arabinan and mannan segments [12, 13]. LAM has been shown and validated to be present in high concentrations in biological fluids such as sputum, serum and urine [14–16]. In recent years, we and many other laboratories have made

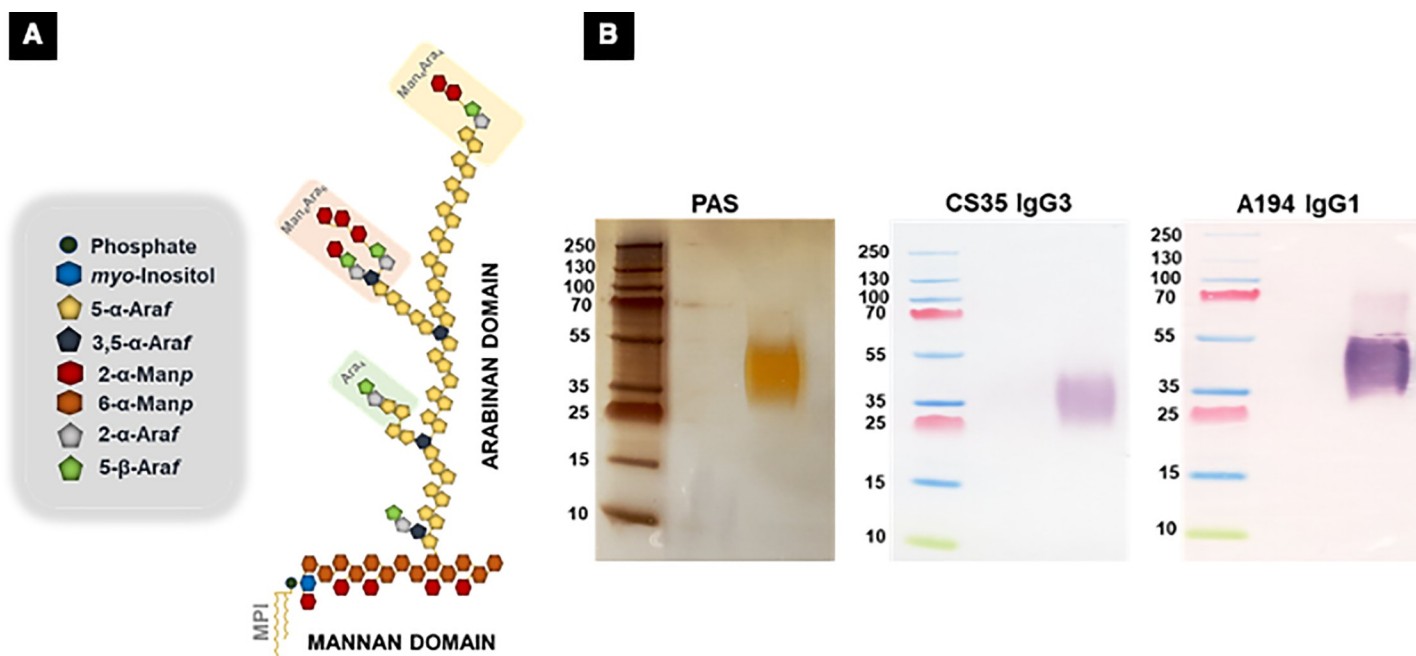

**Fig 1. A.** Structure of LAM. A major component of the cell envelope of Mtb is lipoarabinomannan (LAM). LAM is a complex of D-mannan and D-arabinan attached to a phosphatidyl-*myo*-inositol (PI) moiety that anchors in the mycobacterial cell wall [7]. The D-mannan consist of a highly branched structure with an α-(1→6)-linked mannopyranose (Man*p*) backbone substituted at C-2 by single Man*p* units. The arabinan consists of a linear α-(1→5)-linked Ara*f* backbone punctuated by branching produced by 3,5-*O*-linked α-D-Ara*f* residues. The lateral side chains are organized either as a linear tetra-arabinofuranoside or as a biantenary hexa-arabinofuranosides [13, 38]. These chain terminating arabinan can be further capped with short mannose α-(1→2)-linked oligosaccharides and are epitopes for binding to mAbs used in this study. **B.** Periodic Acid Silver Staining of Mtb CDC1551 LAM. LAM purified from *in vitro* grown cells showing a tight smear (MW ~ 15–17 kDa). Western blot profile of the CDC1551 LAM with the anti-LAM mouse monoclonal CS35 antibody and anti-LAM human monoclonal A194, the two antibodies used as a pair in our Capture ELISA.

significant advances in urinary LAM-based diagnostics for active TB [17–19]. Specimens like serum and urine collection are less invasive and can be easily collected from patients admitted to the hospital or even outside of the hospital setting. A point-of-care (POC) test that readily detects active TB would reduce diagnostic delays, interrupt transmission with appropriate therapy, and address many of the current gaps in global TB control.

Current commercial urinary LAM-based diagnostic kits have a specificity of 95% or more and sensitivity of 40–70% but are recommended only for patients with HIV and CD4 counts less than 100 cells per microliter. One such test, Alere Determine TB LAM Ag (AlereLAM), is a lateral flow assay designed to detect LAM in urine and used as a POC test for active tuberculosis [20–22]. A second test in the market, FujiLAM has been recently developed and validated [18]. This assay involves the use of two high affinity anti TBLAM monoclonal antibodies (mAb) accompanied by a silver amplification step [23] to intensify the control band and signal bands for LAM in urine, thus increasing the sensitivity of the assay compared to the AlereLAM test. We on the other hand, focused on validation of a wide range of clinical samples with a simple capture ELISA (C-ELISA) including a proteinase K (ProK) sample pretreatment [24] to alleviate background and open up epitopes if sequestered. This C-ELISA can be implemented in a clinical setting but not as a POC test.

Paper have been used in analytical chemistry for centuries in the form of spot-tests, litmus paper and lateral flow assays based on nitrocellulose [25–28]. Properties of paper, including low-cost, ease of fabrication, wide availability, and self-wetting, make it an ideal substrate for sensors for the developing world. In addition, properly designed immobilization can be beneficial to almost all enzyme properties, such as activity, specificity, selectivity, reduction of inhibition and aforementioned stability. Nevertheless, the resource limited settings often lack proper refrigeration for storage and transport and can suffer from lack of supplies, batch to batch variation in reagents, and other problems.

ProK is a broad serine protease and cleaves proteins at the carboxyl side of the aromatic and hydrophobic amino acids. The enzyme shows maximum activity in the pH range of 7.0–12.0 and $Ca2+(1.0–5.0mM)$ is required for activation [29]. Additionally, ProK maintains its activity in the presence of detergents often used in an assay. Immobilization of an enzyme increases its durability under harsh conditions and simplifies its removal from the reaction medium before the rest of the assay is completed. The use of immobilized enzymes in combination with ProK is used for the removal of protein impurities from biological fluids where nucleic acid-degrading enzymes (e.g. DNases) need to be eliminated. On the other hand, there are not many published works on the immobilization of ProK.

Our approach towards POC test development was first to immobilize ProK on a Whatman no.1 paper and demonstrate its applicability in the C-ELISA. (**Schematics in Fig 2**)

In this report we show that we were successful in our immobilization of ProK and the immobilized strips (IPK) perform similarly to the standard 'add and deactivate' procedure. Twenty five clinical samples from the Foundation of Innovative New Diagnostic (FIND) were tested to demonstrate the post pretreatment outcome. The assays to monitor efficiency were done with two different ELISAs- a straight forward indirect ELISA to determine optimum conditions [17] and C-ELISA for sample validation [24].

## Materials and methods

### Ethics statement

Anonymized archived urine samples used in this study were provided in 2014 by the Foundation for Innovative New Diagnostics (FIND, Geneva) and stored in Colorado State University (CSU) [30]. The study samples cannot be traced back to individual patients. The study was

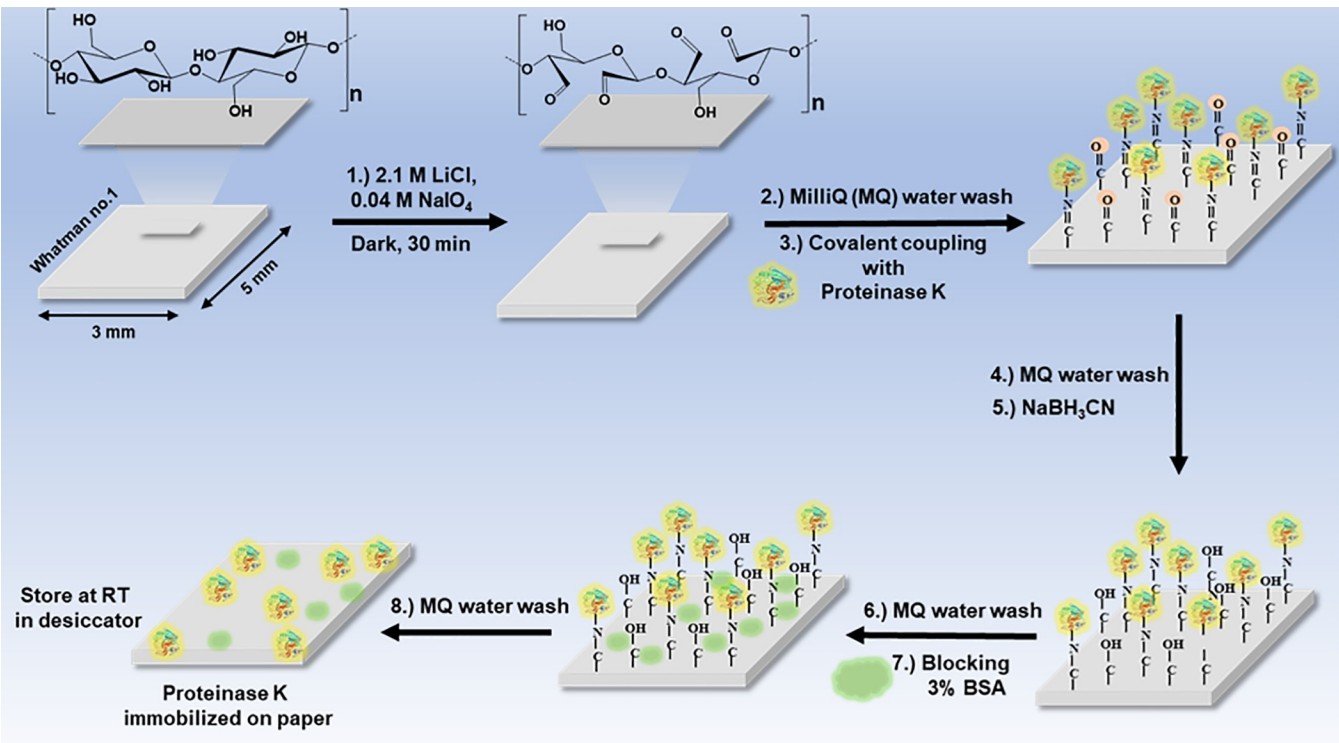

**Fig 2. Scheme of immobilizing Proteinase K (IPK).** Schematic representation of the immobilization of Proteinase K on Whatman paper # 1. Lithium chloride in the presence of sodium Periodate was used to change the functional group of the paper from hydroxyl to aldehyde group. Proteinase K was covalently immobilized on the charged paper and treated with sodium cyanoborohydride to preserve the covalent bonds, followed by blocking the non specific binding sites.

approved by the local IRB and all subjects providing samples signed an informed consent at the time of enrollment and before sample collection. Participants were informed that the samples were going to be stored at FIND repository and will only be used for the development of new TB diagnostics.

None of the authors have access to identifying patient information.

## Clinical sample cohort

The urine samples were collected from patients of both sexes suspected of or showing symptoms of pulmonary TB, with and without HIV co-infection, presenting prior to the initiation of treatment at clinics in Vietnam, South Africa and Peru. The specimens were sedimented by centrifugation immediately after collection and the supernatant was stored at -80˚C within a few hours of collection. The clinical diagnosis of TB vs non-TB was based on sputum smear microscopy plus>2 and sputum cultures in addition to other clinical and radiological examinations.

Patients without a firm final diagnosis (e.g. contaminated culture, persistent symptoms despite repeated negative TB cultures, or treatment for TB without culture-confirmation) were excluded from study.

Anonymized urine control samples were obtained from healthy volunteers from a TB non-endemic region (NEU) in CSU, aliquoted and stored frozen at -80˚C until further use (approved by CSU IRB under approval numbers 15-104B and 09-006B). The control urine was spiked with Mtb CDC1551 LAM (ranging from 0.001 μg/mL– 1 μg/mL for indirect ELISA and (0.02 ng/mL to 12.5 ng/mL for C-ELISA) for optimization of the IPK pretreatment conditions

and to generate an assay standard curve by serially diluting the LAM two-fold to obtain a concentration range in comparison to the unspiked urine which was used as a background negative control.

## Proteinase K immobilized on paper (IPK)

Proteinase K was immobilized on Whatman paper no. 1 via a covalent bond as described in previous reports with some modifications [31, 32]. Whatman paper no. 1 was cut into 3×5 mm strips and 5 μl of 2.10 M lithium chloride in 0.04 M sodium periodate was dropped on the strips to modify the functionality of paper from hydroxyl to aldehyde group and maintained in dark for 30 min. After 30 min of oxidation, treated paper was washed with sterile milliQ water (MQW) X 3 followed by dabbing the excess water on a blotting paper. ProK (dry powder, Thermo Scientific) in 2 μL was immobilized at the required concentration/s (0–1000 μg/mL) onto the modified paper and incubated in dark for 30 min followed by washing with MQW. Sodium cyanoborohydride at 1 mg/mL (5 μL) was added to the paper strips for 5 min to obtain stable covalent bonds (secondary amines) and washed. Subsequently, the immobilized paper strips were blocked by adding 3% BSA for 15 min and washed. The immobilized paper was then dried at 37°C and stored in a desiccator at room temperature (at 27°C in an incubator) until further use.

## LAM for assay standardization

The LAM used in this study was isolated and purified from Mtb CDC1551 strain in *in vitro* culture here at CSU as previously described [33]. LAM was isolated in large quantities so that the same standard could be used throughout the year for recurring experiments.

## Antibodies

A mouse monoclonal antibody CS35 IgG3, raised against *Mycobacterium leprae* LAM was isolated from the hybridoma cell line generated by the fusion of myeloma cells with immunized mouse splenocytes as described previously [24, 34]. No animal experiments were performed for the current study since the hybridoma cells were generated 30 years back and the hybridoma cell line stored in the CSU repository. All IACUC animal protocols were followed during the generation of the antibodies and the animals received proper care using all humane guidelines and institutional regulations of care and euthanization based on the protocols in place at the time by CSU staff veterinarian and the animal care staff. An approved Animal Care and Use Committee (ACUC) protocol and Animal Welfare Assurance number A 3572–01 was on file with the Office of Protection from Research Risks along with the approved mouse protocol for the production of monoclonal antibodies under the former NIH/NIAID Leprosy Research Materials Contract No 1 A1-52582 here at CSU. The records for this specific animal protocol number have been lost during the 30 year time span.

A human mAb, A194 IgG1 was obtained from New Jersey Medical School (Rutgers University). The antibody was molecularly cloned from a patient diagnosed with pulmonary TB who had already started on drug treatment for a month before screening the culture supernatant against ManLAM in an ELISA assay using a high throughput *in vitro* B cell culture method as described elsewhere [35]. Application of these two Abs (CS35 and A194-01) has been described in our previous published work [17, 24].

## Standard ProK pretreatment (SPK)

ProK was added to the urine samples at a final concentration of 200 μg/mL and incubated at 55°C for 30 min followed by inactivation at 100°C for 10 min. The treated samples were then

centrifuged at 12,000 x g for 10 min and the supernatant used for C-ELISA. For indirect ELISA, ProK was used at 200 μg/mL to pretreat the urine spiked with LAM at 55˚C for 2hr followed by inactivation at 100˚C for 30 min. The pretreated samples were then centrifuged at 12,000 x g for 10 min and the supernatant used for the ELISA assay.

## Immobilized ProK pretreatment (IPK)

For optimization of IPK, ProK was immobilized on the Whatman paper # 1 at varying concentrations (ranging from 0 μg / mL– 1000 μg / mL) and tested in an Indirect ELISA platform using 200 μL as sample volume. A time course was setup starting from 0, 30, 60, 120 and 180 mins and pretreatment with IPK performed at room temperature. Once the optimal concentration for IPK was achieved at 400 μg/mL with optimal time between 60 to 120 min, a temperature analysis was performed for 60 mins with 400 μg/mL IPK at room temperature, 37˚C and 55˚C. Best results were obtained at room temperature.

## Indirect ELISA to confirm the immobilization of Proteinase K on paper strip

To optimize the concentration of ProK to be used, the optimal incubation time and temperature for the pretreatment, indirect ELISA (which measures binding of antibody to the antigens) was carried out as previously described [36] with modifications. Urine from a healthy volunteer was spiked with known amounts of LAM (ranging from 0.001 μg/mL– 1 μg/mL) and pretreated with IPK strips and then used for coating a 96-well plate (Corning Costar) in equal volume of the coating buffer (0.05M carbonate bicarbonate buffer, pH 9.6) and the plate incubated at 4˚C overnight. Non-specific binding sites were blocked with 1% bovine serum albumin (BSA) (Sigma Aldrich) in 1x phosphate buffered saline (PBS) (blocking buffer) after washing the wells briefly with the same. Purified CS35 was used at a concentration of 2 μg/mL and added to the wells and incubated for 90 min at room temperature. The plates were then washed with the wash buffer (1XPBS with 0.05% Tween-80) and then incubated for 90 min with anti-mouse IgG alkaline phosphatase conjugate (Sigma) for the murine primary antibody, diluted 1:2500 in wash buffer. The plates were again washed and the alkaline phosphatase activity measured by addition of p-nitrophenyl phosphate (pNPP) (Thermo Scientific) as a substrate. The optical density was measured at 405nm. All standards were run in duplicates and the absorbance plotted to determine the binding activity of the antibody to the LAM. As a control for the IPK pretreatment, NEU spiked with LAM was simultaneously pretreated by addition of ProK at 200 μg/mL final concentration at 55˚C for 2 h followed by inactivation at 100˚C for 30 min.

## Capture ELISA on NEU spiked with LAM and clinical samples

For optimization of the IPK concentration, time of exposure and temperature for pretreatment on a capture ELISA (C-ELISA) platform, previously published protocol was followed with slight modifications [17, 24]. A 96 well polystyrene high binding plate (Corning Costar) was coated with a capture antibody (CS35 ms mAb) at 10 μg/mL concentration in PBS and incubated at 4˚C overnight. NEU spiked with known amount of LAM (ranging from 0.02 ng/mL– 12.5 ng/mL) was incubated at room temperature for 30 min to allow for the complexation of LAM and urine protein/s, followed by storing at -20˚C overnight to somewhat mimic the conditions for the clinical samples. After overnight incubation, the antibody plates and the LAM samples were brought to room temperature and the plates blocked with 1% BSA in 1X PBS (blocking buffer) for 1 hr after briefly washing the plates with the same. Control and/or clinical samples were pretreated with Prok using the SPK method and concurrently the samples were

pretreated with IPK by the addition of the strip into a second sample tube for the required time and the samples then used for ELISA. The plates were washed with the blocking buffer, the control and/ or the clinical samples were added to the appropriate wells and incubated for 90 min at room temperature. The plates were then washed with the wash buffer (1x PBS—0.05% Tween-80) and the biotinylated detection antibody (A194hu IgG1) added at a concentration of 250 ng/mL to all the wells and the plates incubated for 90 min at room temperature. Following another wash, 1:200 dilution of Streptavidin-Horseradish Peroxidase (HRP) (R & D Systems) was added to the plates and incubated for 25 min at room temperature. After the final wash, Ultra TMB chromogenic substrate (Thermo Scientific) was added to all the wells and the plates incubated for at least 30 min and observed for color development. The reaction was stopped by adding 2M Sulphuric Acid (Fisher Scientific) to the wells and the plates read at 450nm.

## Statistical methods

Correlation was evaluated with Spearman's $\rho$. P-values are based on a test of the null hypothesis that correlation is equal to zero. Analyses were conducted in the open software R version 4.0.4 (2021-02-15) using base functions.

## Results and discussion

### Proteinase K immobilization on Whatman no.1 paper (IPK)

In this study we adapted a facile universally applicable method for ProK immobilization. Optimal amounts needed, time to complete digestion, operation temperature, and stability time on the paper were also determined and were monitored by ELISA.

Whatman no.1 paper were excised (3x5 mm) and the -OH groups of the carbohydrates were converted to aldehydes via periodate oxidation using $NaIO_4$ as an oxidizing agent. We used LiCl in order to enhance the periodate oxidation efficiency because it makes hydroxyl groups more available to periodate oxidation. ProK was covalently linked to the aldehyde groups formed on the paper to create a reverse Schiff base [37]. The remaining aldehyde groups and the Schiff base were reduced by sodium cyanoborohydride ($NaCNBH_3$) via reductive amination as depicted in the schematics in **Fig 2**.

### Optimization of Proteinase K on Whatman no.1 paper (IPK)

**Concentration.** To test for the immobilization of ProK on Whatman paper, a BCA assay was performed on the ProK stock tube (0–1000 μg/mL) and the washes from the immobilization steps of strips immobilized with varying concentrations (0–1000 μg /mL) of ProK. All samples were tested in duplicate and plotted against the stock ProK curve. We observed no significant loss of ProK during the immobilization steps (**Fig 3**). These results indicated that the concentration of ProK immobilized on the strips were as specified (0–1000 μg /mL) and no excess ProK washed out. To optimize the concentration of ProK for assay performance, we spiked urine collected from a healthy volunteer with LAM starting at 1μg /mL and used strips containing varying concentrations of the enzyme (0–1000 μg /mL) treated for 2 hr at room temperature and then analyzed using indirect ELISA with CS35 mAb. At concentrations 0 and 50 μg /mL, the $OD_{405}$ values were at or near the background levels. At higher concentrations (100–1000 μg/mL), LAM showed increased binding to the Ab with the lowest background at 400 μg /mL (**Fig 4A**). This also confirms retention of ProK on paper after immobilization. As a comparative control for IPK, non-endemic urine (NEU) spiked with LAM was simultaneously

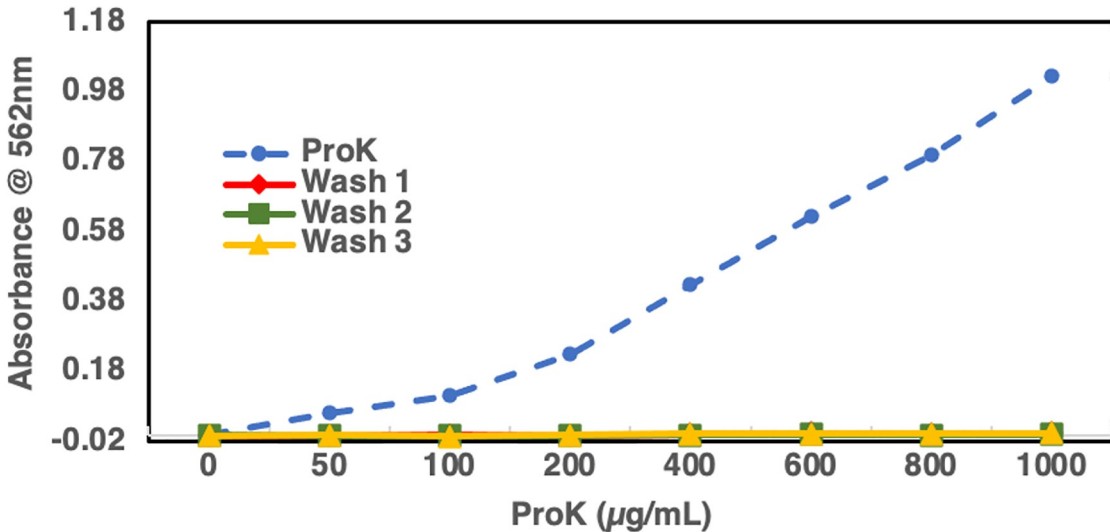

**Fig 3. Graphical representation showing negligible loss of Proteinase K.** During the immobilization steps where Wash 1, 2 and 3 are from the washes tested in a BCA protein estimation assay from steps following the immobilization of Proteinase K to the filter paper, treatment with $NaBH_3CN$ to preserve the covalent bonding and the final wash before the blocking of the non specific sites respectively. The blue dotted line shows the Proteinase K stock solution used for immobilization.

treated with SPK and used for C-ELISA, the $OD_{450}$ values were similar to what we obtained with the IPK treatment (**Fig 4B**).

**Time.** Using 400 μg/mL as the optimal concentration on IPK in an indirect ELISA, we set out to optimize the time of pretreatment required. To achieve this, urine spiked with LAM was treated with IPK at 400 μg /mL at a time interval of 0, 30, 60, 120 and 180mins at room temperature and analyzed by indirect ELISA. We observed the best signal as compared to 0 min (**Fig 4C**) at 60 and 120 min concluding that in indirect ELISA, in urine spiked with LAM, IPK best performs at a concentration of 400 μg/ml for 60 min at room temperature.

**Temperature.** Since SPK is optimized at a higher temperature (55˚C) [24], we needed to optimize the pretreatment temperature for IPK. We pretreated the NEU spiked with LAM with IPK at 400 μg/mL for 60 min at room temperature, 37˚C (ProK can be activated at this temperature) and 55˚C. We observed that at high temperatures (37˚C & 55˚C), $OD_{405}$ improved with concomitant increase in background. At room temperature however, although the absorbance values were lower, there was very low background. This led us to conclude that at 400 μg/mL with IPK, the optimal urine pretreatment could be done at room temperature for 60min (**Fig 4D**). Incidentally, background interference is one of the most critical issues in developing sensitive POC assays for TB diagnosis, as the analyte concentration is low in a majority of the population.

**C-ELISA.** To optimize conditions for the use of IPK in the C-ELISA platform, we pretreated the urine spiked LAM with IPK at various concentrations (0–1000 μg/mL (**Fig 5A**) at room temperature for 60 min. We noted that at 400 μg/mL, as we had observed initially in the indirect ELISA, the $OD_{450}$ values were significantly higher than at 0, 50, 100 and 200 μg/ml, and the background was much lower. We set out to optimize the time and temperature required for the IPK pretreatment that can be used for analyzing clinical samples. We used LAM spiked urine pretreated with IPK at 400 μg/mL for a time course of 0, 10, 30 and 60 min at room temperature and 55˚C. We observed that as compared to the 0 min pretreatment at room temperature and 55˚C, there was no significant difference in the absorbance values at different time points and temperatures (**Fig 5B**). This led us to conclude that for C-ELISA of

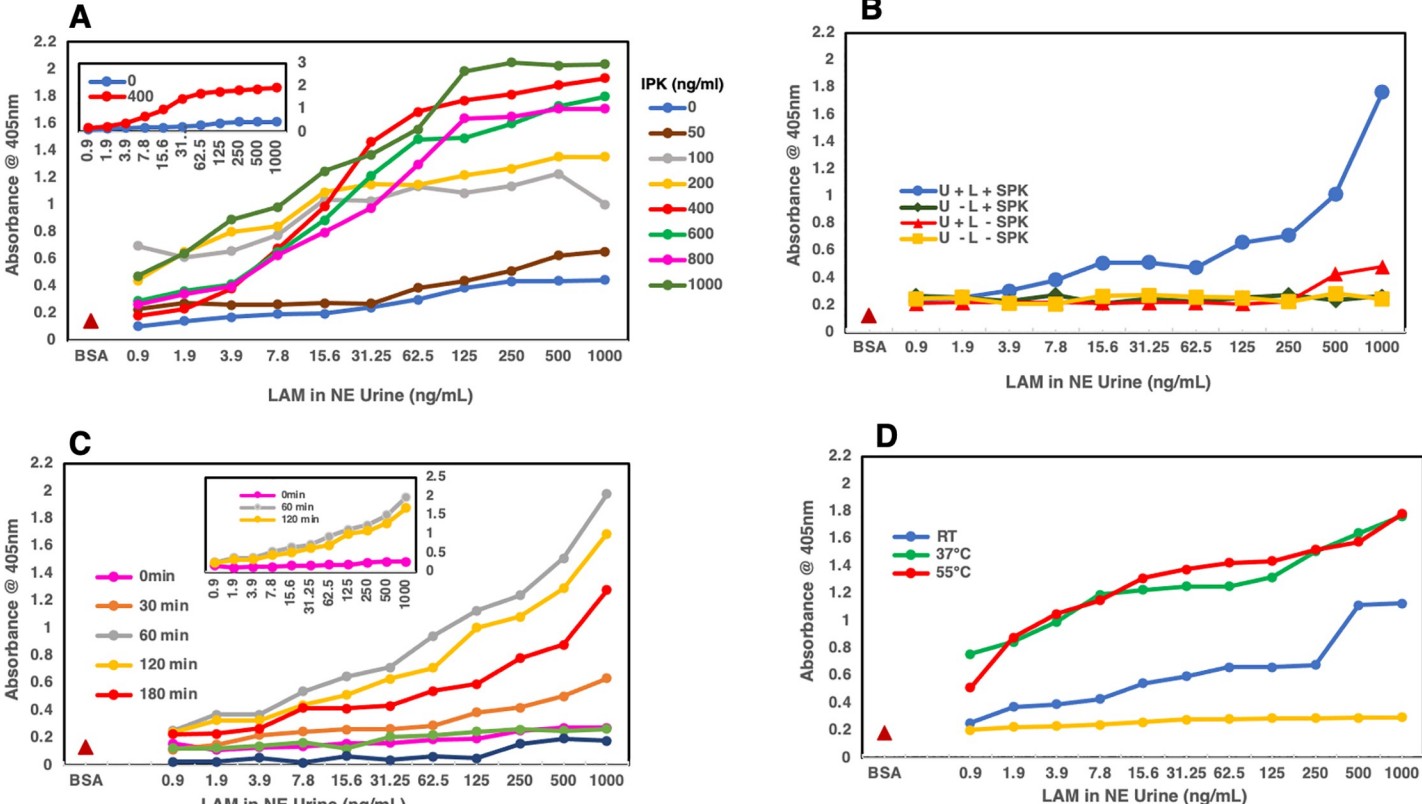

**Fig 4. Indirect ELISA. A)** Shows the concentration curve (0–1000 µg /mL) for IPK that would give optimal results when used for the pretreatment of urine spiked with LAM. The inset shows 400 µg /mL of IPK (red) to be the optimal concentration that gives the best results compared to the urine spiked with LAM that did not see any IPK (dark blue). **B)** Showing the standard Proteinase K (SPK) treatment of urine from a healthy volunteer spiked with LAM purified from *M.tb* CDC1551 cells. The blue line with circles shows the effect of SPK in sequestering LAM from the protein/s or other inhibitors present in urine compared to the red line with triangles where the urine spiked with LAM has not been pretreated with SPK. The yellow line shows the background signal obtained with the urine and the green line with diamonds shows that Proteinase K itself does not have any effect on blank urine. **C)** Shows the optimal time required for the pretreatment, with IPK at 400 µg /mL, of urine spiked with LAM. At 60 min (grey line) and 120 min (yellow line) (see inset) compared to the 0 min (pink line) pretreatment time, more LAM seems to be available for binding to the antibody. **D)** Shows the optimal temperature needed (55˚C) for the IPK pretreatment of urine spiked with LAM for 60 min at 400 µg/mL of IPK.

urine clinical or spiked urine samples using IPK as pretreatment, 400 µg/mL at room temperature for 10–30 min should be optimal to achieve the release of LAM from urine. However, since the surface area of the strips is small (application volume max-5 µL), we suggest a 30 min exposure of one strip per 200 µL sample size is more desirable.

**Clinical samples.** We have shown repeatedly that during assay or method development, clinical samples do not perform in a similar manner to the control urine sample spiked with LAM. To test our newly developed procedure for sample pretreatment (IPK), we analyzed 25 clinically characterized urine samples from TB patients or suspects that had been previously validated using chemical derivatization method developed in our laboratory utilizing gas chromatography/mass spectrometry (GC-MS) [30]. Of these 25 urine samples, 12 were smear and culture positive and 13 were non-TB [30]. The samples were collected about 10 years ago and stored at -80˚C in Colorado State University (CSU) lab repository. These 25 samples (100 µL each) were pretreated simultaneously with SPK and IPK and C-ELISA performed on all. Unlike the control LAM-spike NEU, OD values for IPK were comparable to SPK. Nonetheless, in both methods, the results from 25 urine samples were in agreement with the clinical status (**Table 1**). As expected, $OD_{450}$ values for both IPK and SPK were lowest (between 0.1 and 0.2)

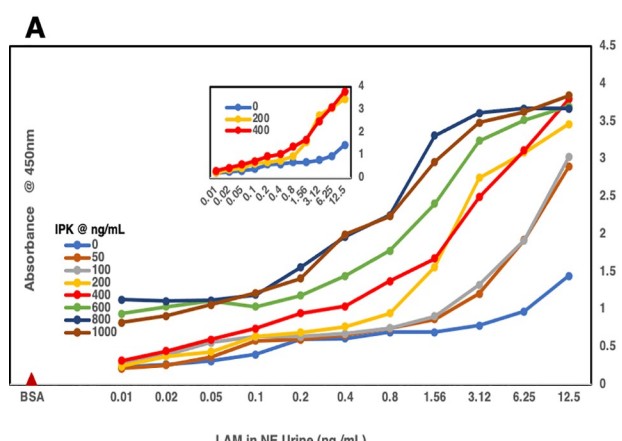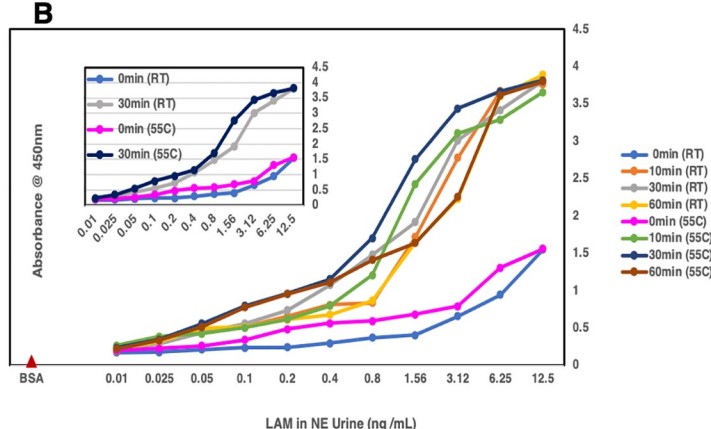

**Fig 5. Capture ELISA. A)** Graph plot showing the concentration curve (0–1000 μg / mL) for IPK pretreatment of NEU spiked with LAM. Optimal results are obtained at 400 μg / mL concentration of IPK as compared to 0 μg /mL (see inset). **B)** Shows the optimal time and temperature required to obtain the best results in a capture ELISA when using IPK for pretreatment of urine spiked with LAM. The graph shows no significant difference between the various time points and temperatures variations, hence the optimal conditions for C-ELISA taken with IPK is 400 μg / mL for 30 min at room temperature.

**Table 1. Twenty five clinically characterized urine samples analyzed in a capture ELISA using both SPK and IPK as the pretreatment for comparison.** A majority of the samples were from Vietnam, included both male and female with ages ranging from 35 to 80.

| Sample ID | Clinical Status | | C-ELISA OD$_{450}$ | | C-ELISA Status |
|---|---|---|---|---|---|
| | Smear | Culture | SPK | IPK | |
| 7 | 2+3+ | C+ | 0.623 | 1.727 | Positive |
| 13 | 2+3+ | C+ | 2.637 | 1.649 | Positive |
| 16 | Non TB | Non TB | 0.253 | 0.272 | Negative |
| 19 | 2+3+ | C+ | 0.451 | 0.469 | Positive |
| 21 | Non TB | Non TB | 0.258 | 0.202 | Negative |
| 25 | 2+3+ | C+ | 0.822 | 0.863 | Positive |
| 35 | Non TB | Non TB | 0.319 | 0.166 | Negative |
| 62 | Non TB | Non TB | 0.146 | 0.252 | Negative |
| 63 | Non TB | Non TB | 0.225 | 0.236 | Negative |
| 99 | 2+3+ | C+ | 3.331 | 1.467 | Positive |
| 112 | Non TB | Non TB | 0.274 | 0.215 | Negative |
| 114 | Non TB | Non TB | 0.243 | 0.203 | Negative |
| 115 | 2+3+ | C+ | 1.056 | 0.575 | Positive |
| 118 | Non TB | Non TB | 0.188 | 0.207 | Negative |
| 123 | Non TB | Non TB | 0.332 | 0.256 | Negative |
| 127 | Non TB | Non TB | 0.276 | 0.196 | Negative |
| 136 | Non TB | Non TB | 0.294 | 0.233 | Negative |
| 160 | Non TB | Non TB | 0.212 | 0.18 | Negative |
| 162 | 2+3+ | C+ | 0.374 | 0.49 | Positive |
| 171 | 2+3+ | C+ | 0.715 | 0.406 | Positive |
| 183 | 2+3+ | C+ | 0.831 | 0.526 | Positive |
| 191 | Non TB | Non TB | 0.219 | 0.257 | Negative |
| 192 | 2+3+ | C+ | 1.597 | 1.4005 | Positive |
| 194 | 2+3+ | C+ | 0.68 | 0.599 | Positive |
| 199 | 2+3+ | C+ | 0.537 | 0.49 | Positive |

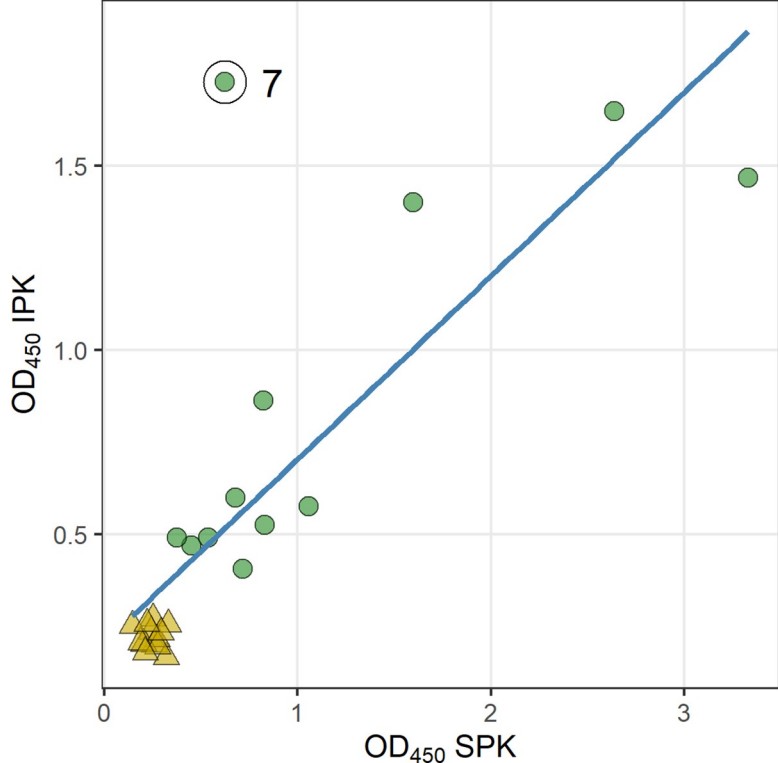

**Fig 6. Correlation of IPK vs. SPK.** Optical density at 450nm for C-ELISA using SPK is plotted on the x-axis; optical density at 450nm for C-ELISA using IPK is plotted on the y-axis, and a regression line is shown ($r^2$ = 0.63). Triangles indicate samples that were non-TB by clinical assignments; circles were TB-positive. The non-TB samples clustered together separately from TB-positive samples in both ELISAs. Sample 7 had an unusually high reading by IPK, as indicated by the circled and labelled point. CS35 IgG is used as the capture mAb and A194-01 IgG1 is used as the detection mAb in the assay.

for the LAM-negative samples. These values clustered together in a scatterplot of IPK vs SPK and were distinctly separate from $OD_{450}$ values for the LAM-positive samples (**Fig 6**). Because the low values group together separately from the higher values, correlation is high: $\rho$ = 0.80, p = < 0.0001, $\tau$ = 0.59, p = < 0.0001). However, it is useful to examine each of the LAM-positive and LAM-negative groups separately. Within the cluster of LAM-negative samples, the OD values are not correlated ($\rho$ = -0.10, p = 0.74; $\tau$ = -0.10, p = 0.68). For the LAM-positive samples, the OD values were widely spread out, and moderately correlated ($\rho$ = 0.57, p = 0.05; $\tau$ = 0.44, p = 0.05). In particular, sample 7 had a much higher $OD_{450}$ value using IPK and was considered as an outlier. The sample set test was done as a initial screen for method feasibility and not a true validation. Our goal is to transition from C-ELISA to paper-based microfluidics devices incorporating IPK when larger validation study will be utilized. With IPK, we have been able to reduce the time taken for the assay from approximately an hour to 30 min. The strips were an improvement over SPK as we did not find unbound enzyme in the wash solutions that would affect the Abs in use and strips can be incorporated into lateral flow devices. The ease of use of IPK cannot be understated. One thing to add will be the stability of LAM in clinical samples over time. Samples tested in this work were collected almost 10 years ago, LAM assay stood well over time. A larger validation study is warranted to determine IPK stability on paper for a period of 6 months to one year and on larger sample set.

## Conclusion

A simple paper based immobilization of ProK has been successfully developed and implemented in clinical urine samples from TB patients. This step is essential for minimizing background associated with detection of LAM in clinical urine samples in a point of care setting. Simplification of the proposed pretreatment strip were achieved by using Whatman no.1 paper and by minimizing the concentration of ProK used to pretreat the clinical samples prior to ELISA. The method is quick, requires minimal amounts of Prok (40 micrograms per 100 microlitres of sample) and causes no leaching. The fascile method is developed keeping in mind a development of a new microfluidic interface platform for on-site enzyme-based LFIA analysis. Further validation with a large set of urine samples is required to determine the merit of this method presented herein.

## Supporting information

**S1 Fig. Periodic acid silver staining of Mtb CDC1551 LAM.** Original raw image uncropped SDS/PAGE and corresponding Western Blot. LAM purified from *in vitro* grown cells showing a tight smear (MW ~ 15–17 kDa). Western blot profile of the CDC1551 LAM with the anti-LAM mouse monoclonal CS35 antibody and anti-LAM human monoclonal A194, the two antibodies used as a pair in our Capture ELISA.
(TIFF)

## Acknowledgments

We gratefully acknowledge Dr. Abraham Pinter from PHRI, NJMS, Rutgers University for the generous contribution of the human mAb A194. The mouse mAb CS35 was obtained from in-house collection at CSU maintained by Dr. John S. Spencer. We gratefully acknowledge the Foundation for Innovative New Diagnostics (FIND) Geneva, Switzerland for providing the clinically characterized patient urine samples. We gratefully acknowledge the critical inputs from Dr. Prithwiraj De (CSU).

## Author Contributions

**Conceptualization:** Yosita Panraksa, Anita G. Amin, Delphi Chatterjee.

**Formal analysis:** Anita G. Amin, Barbara Graham.

**Funding acquisition:** Delphi Chatterjee.

**Investigation:** Charles S. Henry.

**Methodology:** Yosita Panraksa, Anita G. Amin.

**Project administration:** Delphi Chatterjee.

**Supervision:** Delphi Chatterjee.

**Validation:** Anita G. Amin, Barbara Graham.

**Writing – original draft:** Anita G. Amin, Delphi Chatterjee.

**Writing – review & editing:** Yosita Panraksa, Barbara Graham, Charles S. Henry.

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
