## [Decision Letter · Decision Letter 0]

28 Jul 2021

PONE-D-21-18899

Immobilization of Proteinase K for Urine Pretreatment to Improve Diagnostic Accuracy of Active Tuberculosis

PLOS ONE

Dear Dr. Chatterjee,

Thank you for submitting your manuscript to PLOS ONE. After careful consideration, we feel that it has merit but does not fully meet PLOS ONE’s publication criteria as it currently stands. Therefore, we invite you to submit a revised version of the manuscript that addresses the points raised during the review process.

We look forward to receiving your revised manuscript.

Kind regards,

Seyed Ehtesham Hasnain

Academic Editor

PLOS ONE

Journal Requirements:

2. You indicated that you had ethical approval for your study. In your Methods section, please ensure you have also stated whether you obtained consent from the patients included in the study or whether the research ethics committee or IRB specifically waived the need for their consent.

5. We noticed you have some minor occurrence of overlapping text with the following previous publication, which needs to be addressed:

- https://www.sciencedirect.com/science/article/abs/pii/S1472979218300842?via%3Dihub

In your revision ensure you cite all your sources (including your own works), and quote or rephrase any duplicated text outside the methods section. Further consideration is dependent on these concerns being addressed.

Additional Editor Comments:

Major Revision

Reviewers' comments:

Reviewer's Responses to Questions

**Comments to the Author**

1. Is the manuscript technically sound, and do the data support the conclusions?

Reviewer #1: Yes

Reviewer #2: Yes

2. Has the statistical analysis been performed appropriately and rigorously? 

Reviewer #1: I Don't Know

Reviewer #2: Yes

3. Have the authors made all data underlying the findings in their manuscript fully available?

Reviewer #1: Yes

Reviewer #2: Yes

4. Is the manuscript presented in an intelligible fashion and written in standard English?

Reviewer #1: Yes

Reviewer #2: Yes

5. Review Comments to the Author

Reviewer #1: Comments:

Latent and disseminated TB is still big problem in diagnosis as well as in treatment regimen. In case if smear negative and culture negative TB-HIV case especially in comorbid situation this assay may have great advantages at POCT lavel. Present study of Panraksa, et al. was very nicely done using all methodological aspect to find out the better outcome in TB diagnosis with high sensitivity and specificity. Following suggestions may make more impact on this article:

Abstract:

• Abbreviation must require for TB and Mtb upon first appearance in the text.

Method:

• Advise to put this section before Result and Discussion.

• Ethical statement is missing. Should write in sub-heading in method section only.

• Provide figure in high resolution (Fig 2, Fig 4).

Results:

• Conclusion part is missing

Reviewer #2: I would like to appreciate efforts of the authors to improve Lipoarabinomannan (LAM) assay for the diagnosis of TB. It is beneficial as it is done in non invasive samples (urine) and it should be promoted with improved sensitivity using various pretreatment methods as done in this study.

6. PLOS authors have the option to publish the peer review history of their article (what does this mean?). If published, this will include your full peer review and any attached files.

Reviewer #1: No

Reviewer #2: No

---

## [Author Response · Author response to Decision Letter 0]

17 Aug 2021

August 13th, 2021

PONE-D-21-18899

Immobilization of Proteinase K for Urine Pretreatment to Improve Diagnostic Accuracy of Active Tuberculosis

PLOS ONE

.

To

Professor Seyed Ehtesham Hasnain

Academic Editor

PLOS ONE 

Thank you for thoroughly reviewing our manuscript. Below, we have now addressed all your queries and listed these point-by point. All our responses (point by point) are hilited in red and copied below.

The two reviewers were appreciative of the work presented and had very constructive suggestions, which we addressed to the best of our ability. Original SDS/PAGE Western Blot without any edits has been included as a Supplementary material S1 Fig. 

Figs 2 and 4 in the manuscript have been replaced with High Resolution Tif Figs. Most importantly. we have included a separate Ethics statement and revised some write ups in the Introduction and Methods and material section. Unfortunately, with me being the corresponding author, the style of writing repeats over time with the usage of similar statements. The edits are clearly visible in the version with tracking.

Hopefully the. resubmission will meet with the Journal style and goals.

Kind regards,

Delphi Chatterjee

Journal Requirements:

Level 1 subheadings corrected in the resubmission 

2. You indicated that you had ethical approval for your study. In your Methods section, please ensure you have also stated whether you obtained consent from the patients included in the study or whether the research ethics committee or IRB specifically waived the need for their consent.

Ethics Statement is elaborated and added in the Methods and material section. Consent forms were distributed by FIND at the time of sample collection in ~2010.

This was an oversight on my part. All our data (clinical details of samples and assay performance) are all included in the manuscript . There are no additional data to provide. We have added a statement as 

Data Availability

All relevant data are presented within the paper. 

There is only one SDS/PAGE Western Blot in the manuscript that is Fig 1B. We have now included the original run in the Supporting Information as S1B Fig. with same Fig Legend. 

5. We noticed you have some minor occurrence of overlapping text with the following previous publication, which needs to be addressed:

- https://www.sciencedirect.com/science/article/abs/pii/S1472979218300842?via%3Dihub

In your revision ensure you cite all your sources (including your own works), and quote or rephrase any duplicated text outside the methods section. Further consideration is dependent on these concerns being addressed.

We have rephrased and cited sections that were similar to the above mentioned paper to the best of our knowledge. The clinical sample cohort section in the Methods and material has been rewritten and indicated in the tracks. The Ab section has also been rephrased

Additional Editor Comments:

Major Revision

Reviewers' comments:

Reviewer's Responses to Questions

5. Review Comments to the Author

Reviewer #1: Comments:

Latent and disseminated TB is still big problem in diagnosis as well as in treatment regimen. In case if smear negative and culture negative TB-HIV case especially in comorbid situation this assay may have great advantages at POCT lavel. Present study of Panraksa, et al. was very nicely done using all methodological aspect to find out the better outcome in TB diagnosis with high sensitivity and specificity. Following suggestions may make more impact on this article:

Thank you for your encouraging comments

Abstract:

• Abbreviation must require for TB and Mtb upon first appearance in the text.

Corrected and changed accordingly

Method:

• Advise to put this section before Result and Discussion.- Done in the resubmitted version

• Ethical statement is missing. Should write in sub-heading in method section only.-Done and introduced Ethics Statement as a sub heading in Methods and material section see p. 7

• Provide figure in high resolution (Fig 2, Fig 4).

Replaced Fig 2 and 4 with High Resolution tif files. We have also used PACE per PLOS ONE guidance to check the submitted Figs quality. Fig 2 has been redrawn.

Results:

• Conclusion part is missing

Conclusion section is now added, see p 18.

Reviewer #2: I would like to appreciate efforts of the authors to improve Lipoarabinomannan (LAM) assay for the diagnosis of TB. It is beneficial as it is done in non invasive samples (urine) and it should be promoted with improved sensitivity using various pretreatment methods as done in this study.

Thank you 

6. PLOS authors have the option to publish the peer review history of their article (what does this mean?). If published, this will include your full peer review and any attached files.

Do you want your identity to be public for this peer review? For information about this choice, including consent withdrawal, please see our Privacy Policy.

Reviewer #1: No

Reviewer #2: No

---

## [Decision Letter · Decision Letter 1]

31 Aug 2021

PONE-D-21-18899R1

Immobilization of Proteinase K for Urine Pretreatment to Improve Diagnostic Accuracy of Active Tuberculosis

PLOS ONE

Dear Dr. Chatterjee,

Thank you for submitting your manuscript to PLOS ONE. After careful consideration, we feel that it has merit but does not fully meet PLOS ONE’s publication criteria as it currently stands. Therefore, we invite you to submit a revised version of the manuscript that addresses the points raised during the review process.

We look forward to receiving your revised manuscript.

Kind regards,

Seyed Ehtesham Hasnain

Academic Editor

PLOS ONE

Journal Requirements:

Additional Editor Comments (if provided):

I have gone through the revised manuscript and also the Authors response to the comments of the reviewers. The manuscript was sent for Major revision and Authors have modified the manuscript keeping in mind the comments of the Reviewers but few minor corrections are required in the introduction section as mentioned by the reviewer 1 which needs to be addressed before the publication of this manuscript.

- Abbreviation must be expanded for TB (first paragraph), Mtb (second paragraph) and LAM (third paragraph) upon first appearance in the text (apart from abstract section).

- Required single abbreviation of Mycobacterium tuberculosis, either Mtb or M.tb.

In my view, the authors have satisfactorily addressed all other comments made by the reviewers and have revised the manuscript accordingly. I recommend this manuscript for minor revision.

Reviewers' comments:

Reviewer's Responses to Questions

**Comments to the Author**

1. If the authors have adequately addressed your comments raised in a previous round of review and you feel that this manuscript is now acceptable for publication, you may indicate that here to bypass the “Comments to the Author” section, enter your conflict of interest statement in the “Confidential to Editor” section, and submit your "Accept" recommendation.

Reviewer #1: All comments have been addressed

Reviewer #2: All comments have been addressed

2. Is the manuscript technically sound, and do the data support the conclusions?

Reviewer #1: Yes

Reviewer #2: Yes

3. Has the statistical analysis been performed appropriately and rigorously? 

Reviewer #1: Yes

Reviewer #2: Yes

4. Have the authors made all data underlying the findings in their manuscript fully available?

Reviewer #1: Yes

Reviewer #2: Yes

5. Is the manuscript presented in an intelligible fashion and written in standard English?

Reviewer #1: Yes

Reviewer #2: (No Response)

6. Review Comments to the Author

Reviewer #1: All corrections have been done as according to comments.

Few minor corrections required in Introduction section only:

• Abbreviation must require for TB (first paragraph), Mtb (second paragraph) and LAM (third paragraph) upon first appearance in the text (apart from abstract section).

• Required single abbreviate of Mycobacterium tuberculosis, either Mtb or M.tb.

Reviewer #2: (No Response)

7. PLOS authors have the option to publish the peer review history of their article (what does this mean?). If published, this will include your full peer review and any attached files.

Reviewer #1: No

Reviewer #2: No

---

## [Author Response · Author response to Decision Letter 1]

1 Sep 2021

PONE-D-21-18899R1

Immobilization of Proteinase K for Urine Pretreatment to Improve Diagnostic Accuracy of Active Tuberculosis

PLOS ONE

Dear Dr. Seyed Ehtesham Hasnain

Academic Editor

PLOS ONE

After going through yours and reviewer’s minor editorial comments, we corrected the following in the main text. We have not made any changes in our financial disclosure.

We have included the following items when submitting the revised manuscript: PONE-D-21-18899R1

• A rebuttal letter that responds to each point raised by the academic editor and reviewer(s). Y uploaded this letter as a separate file labeled 'Response to Reviewers'.

With Regards,

Delphi Chatterjee

• 

Additional Editor Comments (if provided):

I have gone through the revised manuscript and also the Authors response to the comments of the reviewers. The manuscript was sent for Major revision and Authors have modified the manuscript keeping in mind the comments of the Reviewers but few minor corrections are required in the introduction section as mentioned by the reviewer 1 which needs to be addressed before the publication of this manuscript.

- Abbreviation must be expanded for TB (first paragraph), Mtb (second paragraph) and LAM (third paragraph) upon first appearance in the text (apart from abstract section)

We agree and this is taken care of in the Introduction

- Required single abbreviation of Mycobacterium tuberculosis, either Mtb or M.tb.

Agreed and this is done

In my view, the authors have satisfactorily addressed all other comments made by the reviewers and have revised the manuscript accordingly. I recommend this manuscript for minor revision.

Minor revisions edited as reconneded

Reviewers' comments:

Reviewer's Responses to Questions

Comments to the Author

1. If the authors have adequately addressed your comments raised in a previous round of review and you feel that this manuscript is now acceptable for publication, you may indicate that here to bypass the “Comments to the Author” section, enter your conflict of interest statement in the “Confidential to Editor” section, and submit your "Accept" recommendation.

Reviewer #1: All comments have been addressed

Reviewer #2: All comments have been addressed

2. Is the manuscript technically sound, and do the data support the conclusions?

Reviewer #1: Yes

Reviewer #2: Yes

3. Has the statistical analysis been performed appropriately and rigorously? 

Reviewer #1: Yes

Reviewer #2: Yes

4. Have the authors made all data underlying the findings in their manuscript fully available?

Reviewer #1: Yes

Reviewer #2: Yes

5. Is the manuscript presented in an intelligible fashion and written in standard English?

Reviewer #1: Yes

Reviewer #2: (No Response)

6. Review Comments to the Author

Reviewer #1: All corrections have been done as according to comments.

Few minor corrections required in Introduction section only:

• Abbreviation must require for TB (first paragraph), Mtb (second paragraph) and LAM (third paragraph) upon first appearance in the text (apart from abstract section).

• Required single abbreviate of Mycobacterium tuberculosis, either Mtb or M.tb.

Reviewer #2: (No Response)

7. PLOS authors have the option to publish the peer review history of their article (what does this mean?). If published, this will include your full peer review and any attached files.

Do you want your identity to be public for this peer review? For information about this choice, including consent withdrawal, please see our Privacy Policy.

Reviewer #1: No

Reviewer #2: No

 PACE used and Figs checked

---

## [Editor Report · Decision Letter 2]

6 Sep 2021

Immobilization of Proteinase K for Urine Pretreatment to Improve Diagnostic Accuracy of Active Tuberculosis

PONE-D-21-18899R2

Dear Dr. Chatterjee,

We’re pleased to inform you that your manuscript has been judged scientifically suitable for publication and will be formally accepted for publication once it meets all outstanding technical requirements.

Kind regards,

Seyed Ehtesham Hasnain

Academic Editor

PLOS ONE

Additional Editor Comments (optional):

I have gone through the revised manuscript and also the Authors response to the comments of the reviewers. The manuscript was sent for Minor revision and Authors have addressed the minor corrections which were required in the introduction section as mentioned by the reviewer 1. I recommend this manuscript for publication.
---

## [Editor Report · Acceptance letter]

10 Sep 2021

PONE-D-21-18899R2 

Immobilization of Proteinase K for Urine Pretreatment to Improve Diagnostic Accuracy of Active Tuberculosis 

Dear Dr. Chatterjee:

I'm pleased to inform you that your manuscript has been deemed suitable for publication in PLOS ONE. Congratulations! Your manuscript is now with our production department. 

Kind regards, 

on behalf of

Prof. Seyed Ehtesham Hasnain 

Academic Editor

PLOS ONE